# Mining latent labels for imbalance classification: a regrouping perspective

## Abstract

Deep learning-based models are sensitive to class imbalance. Existing approaches often involve rebalancing tricks such as loss reweighting and class resampling to emphasize the minority class. In this work, we explore a novel baseline method to deal with class imbalance by regrouping the majority class into smaller pseudo-classes and turning the imbalanced classification problem into a balanced multi-class classification. This simple modification helps to make the class frequencies more uniform in the training data and, simultaneously, helps the representation learning by imposing a structure on the majority class. Experiment results on binary and multiclass classification show that the proposed method can substantially boost the classification performance as measured by average precision metric. Our code will be released before publication.

## 1 Introduction

Modern data classification is founded on accuracy maximization or equivalently error minimization:

$$\min_{f \in \mathcal{H}} \mathbb{E}_{(\boldsymbol{x},y) \sim \mathcal{D}_{\boldsymbol{x},y}} \mathbf{1}\left\{y \neq f(\boldsymbol{x})\right\} = \sum\nolimits_{i=1}^{C} \mathbb{P}(y=i)\mathbb{E}_{\boldsymbol{x}|y=i}\mathbf{1}\left\{y \neq f(\boldsymbol{x})\right\}, \qquad (1)$$

where $\mathcal{H}$ is the hypothesis class, $\mathcal{D}_{\boldsymbol{x},y}$ is the distribution of the input-label pair $(\boldsymbol{x}, y)$, and $C \geq 2$ is the number of classes. Imbalanced Classification (IC) refers to the scenario in which the label distribution $\mathcal{D}(\boldsymbol{y})$ is non-uniform. This has been observed in numerous real-life application tasks, such as rare disease diagnosis (Codella et al., 2018; Irvin et al., 2019), insurance fraud detection (Wei et al., 2013; Herland et al., 2018), object detection (Lin et al., 2017; Chen et al., 2020), text classification (Fernández et al., 2018; Wei & Zou, 2019). For IC, accuracy $\sum_{i=1}^{C} \mathbb{P}(y=i)\mathbb{E}_{\boldsymbol{x}|y=i}\mathbf{1}\left\{y = f(\boldsymbol{x})\right\}$ as the evaluation metric is known to be biased, as it puts more emphasis on classes with high frequencies and hence is less sensitive to performance on minority classes. A slightly refined notion is balanced-accuracy (BA) $1/C \cdot \sum_{i=1}^{C} \mathbb{E}_{\boldsymbol{x}|y=i}\mathbf{1}\left\{y = f(\boldsymbol{x})\right\}$, which cancels out the effect of class frequencies and leads to balanced-error minimization:

$$\min_{f \in \mathcal{H}} \quad 1/C \cdot \sum\nolimits_{i=1}^{C} \mathbb{E}_{\boldsymbol{x}|y=i}\mathbf{1}\left\{\boldsymbol{y} \neq f(\boldsymbol{x})\right\}. \qquad (2)$$

Balanced error minimization forms the basis for two classical ideas to deal with imbalanced classification: **(1) loss reweighting**: assigning larger weights to losses from minority classes compared to majority classes to ensure that the classifier does not bias toward the majority classes (Elkan, 2001); **(2) class resampling** oversampling the minority classes and/or downsampling the majority classes to rebalance the classes. Randomly oversampling (ROS) minority classes is more effective and hence popular than downsampling majority classes (Buda et al., 2018)—throwing away data seems unwise for training modern classifiers such as data-hungry deep neural networks (DNNs). These methods are overwhelmingly popular in practice due to their relative simplicity, flexibility in terms of adapting to different classification models, and their asymptotic guarantees (Menon et al., 2013; Singh & Khim, 2022). However, it should

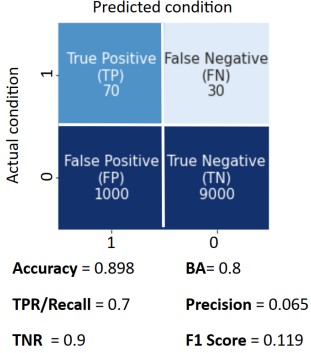

Predicted condition

|  | | |
|---|---|---|
| | True Positive (TP) 70 | False Negative (FN) 30 |
| | False Positive (FP) 1000 | True Negative (TN) 9000 |

Accuracy = 0.898    BA = 0.8

TPR/Recall = 0.7    Precision = 0.065

TNR = 0.9    F1 Score = 0.119

Figure 1: An example confusion matrix for binary IC.

be noted that BA does not provide a complete picture: a classifier with a reasonably large BA may have very poor precision as shown in Fig. 1. Moreover, recent studies (Byrd & Lipton, 2019; Xu et al., 2021) find that the effect of reweighting vanishes as powerful DNNs gradually overfit the training data, nullifying the asymptotic guarantees.

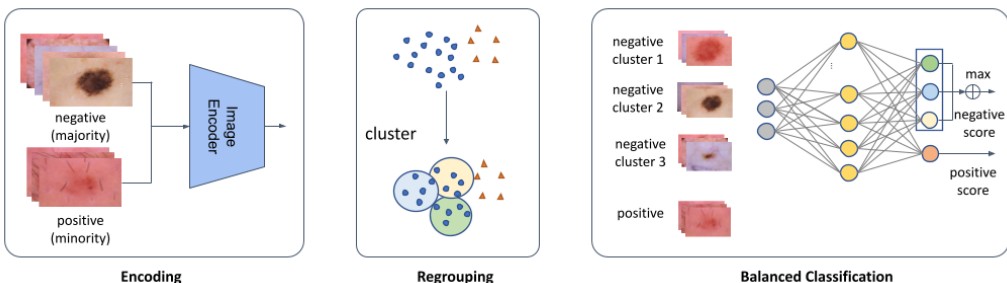

Figure 2: An overview of the training pipeline for the proposed regrouping-based IC method for images. There are three major steps: **(1) Encoding.** Map the input images to a lower-dimensional feature space using a pre-trained model, such as CLIP image encoder; **(2) Regrouping.** Apply clustering methods on majority classes and assign pseudo-labels to the resulting clusters; **(3) Balanced Classification.** Train a classification model with the new balanced dataset.

In this paper, we introduce a fundamental yet relatively under-explored approach to address the issue of class imbalance in classification tasks. Our method involves regrouping the majority class into multiple pseudo-classes, thereby inducing a balanced classification task. It is motivated by the observation that class imbalance often arises artificially due to the labeling process. For instance, in the medical domain, negative controls, which constitute the dominant class, are sourced from various origins or demographic groups. By regrouping these negative controls into smaller, semantically meaningful clusters, we are able to obtain more fine-grained labels leading to relatively more balanced multiclass datasets. While Wu et al. (2010) also explores the use of clustering to decompose the majority class into sub-classes, their primary motivation aligns with the resampling approach, which only aims to balance label frequencies. Additionally, they limit their scope to linear classifiers applied to structured tabular data. In contrast, our work departs from prior approaches. We focus on discovering the latent label hierarchy within the majority class and study image classification with DNNs. The entire pipeline of our proposed methods is illustrated Fig. 2. Importantly, our approach applies to high-dimensional image data by integrating with image encoders, such as CLIP (Radford et al., 2021), or alternative image encoders trained using self-supervised learning techniques.

Our contributions are threefold: 1) We present a novel approach for addressing class imbalance, dubbed as **Re**Grouping (RG), which serves as a strong baseline in addition to class resampling and loss reweighting; 2) We conduct a comprehensive analysis of the proposed regrouping method, demonstrating its ability to facilitate learning efficient representation and synchronizing the training progress across different classes; and 3) We perform experiments on both binary and multiclass IC tasks, demonstrating superior performance compared to state-of-the-art methods in terms of balanced accuracy (BA) and average precision (AP) metrics.

## 2 REGROUPING FOR IMBALANCED CLASSIFICATION

### 2.1 MOTIVATING EXAMPLE

Consider a synthetic imbalanced classification problem: classifying the images in the CIFAR-10 into airplane vs the rest of the 9 classes (such as automobile, bird, ship, etc.). Since the original dataset is balanced in 10 classes, the resulting binary classification problem is imbalanced with a ratio of 9:1. There are two natural approaches to training a classifier for this task: 1) **Binary**: train a binary classifier using 0/1 labels; 2) **Multiclass**: train a multiclass classifier with actual 0-9 labels and aggregate the responses to binary labels at test time. Note that the multiclass approach has two advantages over the binary: 1) the learning problem is balanced; 2) it can leverage extra information in the form of fine-grained labels. However, for a practical imbalanced problem, fine-grained labels

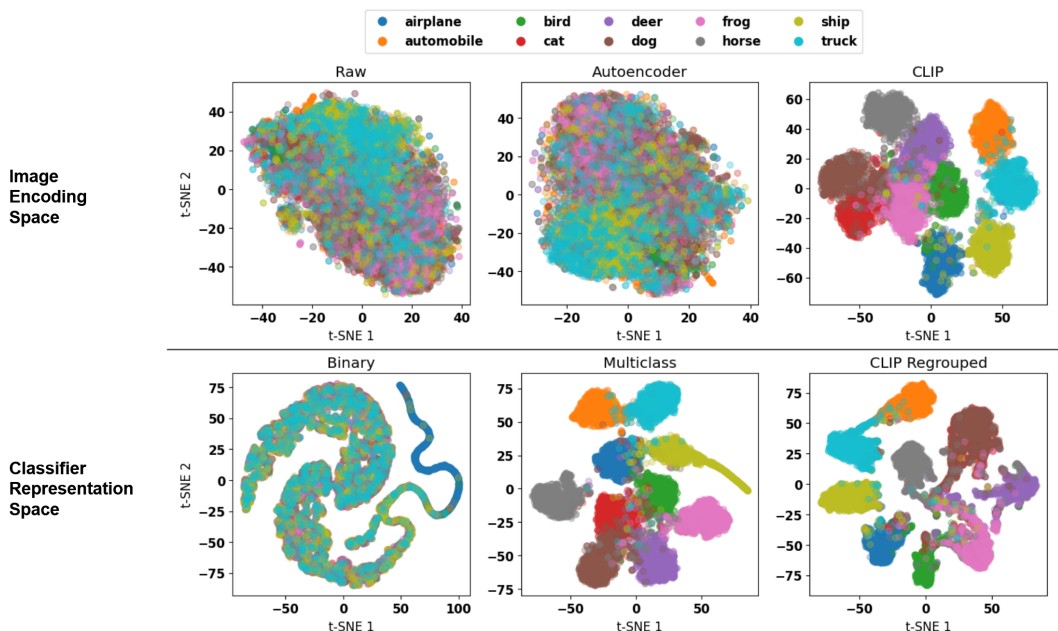

Figure 3: **Top**: t-SNE visualization for CIFAR-10 test set in different image encoder representation spaces; **Bottom**: t-SNE visualization for CIFAR-10 test set classifier representations, i.e. the outputs from the second-last layer of ResNet34 model, trained using different types of labels.

may not be available. A natural idea in this scenario is to regroup the majority class samples into several pseudo-classes using a clustering algorithm.

When the input data is in tabular form, K-means algorithm with Euclidean distance metric is a popular choice for clustering algorithms. However, when the input is in the form of unstructured data such as images, text, or audio, Euclidean distance fails to provide any meaningful information. To illustrate this, in Fig. 3 (Top) we show the t-SNE visualization of CIFAR-10 test data encoded using three approaches: 1) **Raw**: in the flattened image domain, 2) **Autoencoder**: in the latent feature space obtained from an autoencoder trained on CIFAR-10, 3) **CLIP**: in the representation space of the CLIP image encoder (Radford et al., 2021). We note that CLIP model, which was pre-trained using 400 million image-text pairs in a self-supervised manner, is able to group the data in a meaningful manner which can be leveraged to cluster the majority class samples. Raw and autoencoder approaches fail to provide meaningful representations to perform clustering, despite autoencoder providing reasonably good reconstructions, see Fig. 4 for reconstructed images. Hence, CLIP image encoder is a natural choice for regrouping. We highlight two key features motivating the regrouping method:

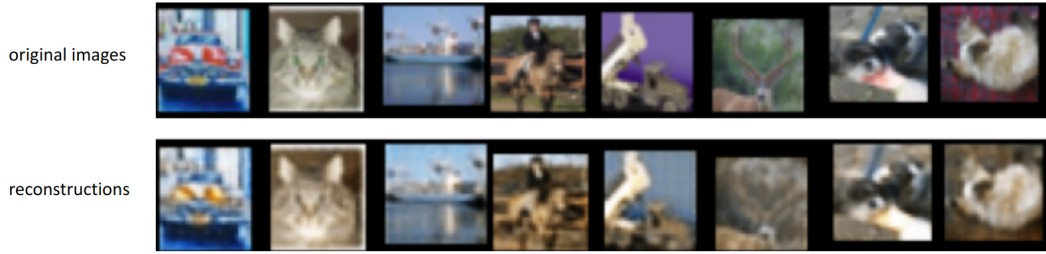

Figure 4: **Top**: original images in CIFAR-10; **Bottom**: reconstruction image from autoencoder.

**Efficient representation learning** For CIFAR-10 training dataset, we obtain the CLIP image encoder representations and use them to cluster the majority class (all images other than airplane class)

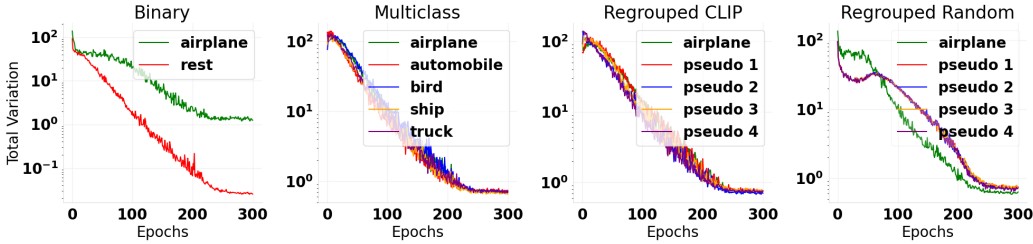

Figure 5: Classifying airplane images vs the rest on CIFAR-10 dataset: The total variation of training sample representations, measured in terms of mean squared distance from the class centers, is compared over epochs.

into 9 pseudo-classes using K-means clustering algorithm. Next, we train a ResNet34 model for the task of airplane vs rest classification using four sets of labels: binary, multiclass (ground truth), CLIP regrouped (airplane and 9 pseudo-classes based on clustering), and randomly regrouped (airplane and 9 pseudo-classes divided randomly). We observe that CLIP regrouped classifier which only uses binary label information is able to learn meaningful feature representation learning (Fig. 3 (Bottom)) similar to multiclass labels, and is able to achieve better test performance (Fig. 6) compared to binary labels. We also note that improved performance is not simply due to balancing the classes as the randomly regrouped labels perform much worse (Fig. 6). We compare the test performance in terms of average precision (AP) metric, which is a standard measure for comparing the performance of classifiers in the presence of data imbalance and is discussed in detail in Sec. 4.

**Synchronous learning**  Apart from semantically separating the majority class samples in representation space, the CLIP regrouped labels are able to to the synchronize the learning pace of different classes. Deep learning models are powerful enough to fit the imbalanced data perfectly, even with binary labels that ignore the class imbalance. However, during the training process, representations of most majority class samples tend to converge close to a single point very rapidly in comparison to the minority class samples. It can be observed in Fig. 5 that the concentration in the case of binary labels is significantly different for majority and minority classes, whereas for multiclass labels, each class tend to concentrate at a similar rate. We refer to this phenomenon as asynchronous learning, where in the initial stage of training the learning is guided by the majority class and in the latter stage by the minority class. CLIP regrouping is able to avoid this issue by learning all classes synchronously, whereas random regrouping is disparate for the airplane class and other pseudo classes. These observations highlight yet another strength of the proposed regrouping method.

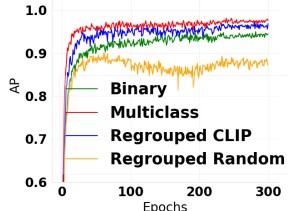

Figure 6: CIFAR-10 test data performance of different methods in terms of average precision (AP) over training epochs

## 2.2 REGROUPING FORMULATION

In this section, we explicitly present the framework of the proposed method. For simplicity, we first present the proposed method in binary classification framework: consider training data $\{(\boldsymbol{x}_i, y_i)\}_{i=1}^n \sim \mathcal{D}_{\boldsymbol{x},y}$, where each input $\boldsymbol{x}_i \in \mathbb{R}^d$ has a corresponding binary label $y_i \in \{-1, +1\}$. The learning model is a DNN $f_{\boldsymbol{\theta}} : \mathbb{R}^d \to [0, 1]$, parametrized by weights $\boldsymbol{\theta}$ and outputting a score for the positive class. In the standard empirical risk minimization (ERM) framework, the weights $\boldsymbol{\theta}$ are trained by:

$$\min_{\boldsymbol{\theta}} \quad \frac{1}{n} \sum_{i=1}^n \ell(y_i, f_{\theta}(\boldsymbol{x}_i)). \tag{3}$$

Without loss of generality, we assume the positive class is rare, i.e., $\mathcal{D}(y = +1) \doteq p < 1/2$, and refer to it as the minority class and the negative class as the majority class. The ERM objective can

then be decomposed as

$$\frac{1}{n}\sum_{i:y_i=-1}\ell(-1, f_{\boldsymbol{\theta}}(\boldsymbol{x}_i)) + \frac{1}{n}\sum_{i:y_i=+1}\ell(+1, f_{\boldsymbol{\theta}}(\boldsymbol{x}_i)). \tag{4}$$

It can be observed that for IC problems with high imbalance ratios (i.e., $(1-p)/p$), the loss in Eq. (4) is dominated by samples in the majority/negative class, i.e., the first summation term. While the typical methods to address the imbalance issue, such as reweighting and resampling, try to rebalance the two summation terms in Eq. (4), we propose to decompose the first term further by assigning pseudo-labels to the negative samples and performing multiclass classification so that the resulting classification problem is more balanced. To this end, we divide the negative samples into $K$ clusters using any standard clustering algorithm (such as K-means), assigning samples belonging to each cluster a pseudo-label $\tilde{y}_i$ from the set $\{2, 3, \dots, K, K+1\}$. The positive samples are assigned the pseudo label $\tilde{y}_i = 1$, resulting in a multiclass problem with $(K+1)$ classes. A classifier (for eg. neural network) $\tilde{f}_{\boldsymbol{\theta}} : \mathbb{R}^d \to [0,1]^{K+1}$ is then trained using ERM:

$$\min_{\boldsymbol{\theta}} \frac{1}{n}\sum_{i=1}^{n}\ell\Big(\tilde{y}_i, \tilde{f}_{\boldsymbol{\theta}}(\boldsymbol{x}_i)\Big) = \frac{1}{n}\sum_{k=1}^{K+1}\sum_{i:y_i=k}\ell\Big(k, \tilde{f}_{\boldsymbol{\theta}}(\boldsymbol{x}_i)\Big),$$

where $\tilde{f}_{\boldsymbol{\theta}}(\boldsymbol{x}) \in [0,1]^{K+1}$ estimate the conditional class probabilities for each pseudo-label. At test time, the decision of binary classification for a sample $\boldsymbol{x}$ can be made using two approaches:

1. **Max aggregation**: classify $\boldsymbol{x}$ to the positive class if $\boldsymbol{e}_1^{\mathsf{T}}\tilde{f}_{\boldsymbol{\theta}}(\boldsymbol{x}) \geq \max_{k\geq 2}\boldsymbol{e}_k^{\mathsf{T}}\tilde{f}_{\boldsymbol{\theta}}(\boldsymbol{x})$,

2. **Sum aggregation**: classify to the positive class if $\boldsymbol{e}_1^{\mathsf{T}}\tilde{f}_{\boldsymbol{\theta}}(\boldsymbol{x}) \geq \sum_{k=2}^{K+1}\boldsymbol{e}_k^{\mathsf{T}}\tilde{f}_{\boldsymbol{\theta}}(\boldsymbol{x})$,

where $\boldsymbol{e}_k$'s are the standard basis vectors. Max aggregation emulates the decision rule typically used in multiclass classification scenario, whereas sum aggregation first aggregates the pseduo classes into actual class probabilities first before comparison. A natural choice for the number of pseudo-labels parameter $K$ is given by the imbalance ratio $n_-/n_+$, where $n_+$, $n_-$ are number of samples in positive and negative classes respectively. However, when the original problem has a very high imbalance ratio, the number of classes may get large. In practice, $K$ may be used as a tunable hyperparameter for optimal performance.

The proposed method can be naturally extended to multiclass classification scenario: Consider a $C$-class imbalanced classification problem: with training data $(\boldsymbol{x}_i, y_i)_{i=1}^{n} \sim \mathcal{D}_{\boldsymbol{x},y}$ where the labels $y_i \in \{1, \dots, C\}$. We assume that the class frequencies $n_j = |\{i : y_i = j\}|$ are sorted in descending order $n_1 \geq \dots \geq n_C$. Similar to the binary case, we aim to tackle the data imbalance by regrouping the samples from classes with large frequencies and assigning them pseudo-labels. To this end, we choose a class frequency threshold parameter $n'$ and regroup the samples from the class $j$ into $\lceil \frac{n_j}{n'} \rceil$ number of clusters respectively, resulting in a K-class classification problem, where $K = \sum_{j=1}^{C}\lceil \frac{n_j}{n'} \rceil$. Note that only the samples from classes frequency $n_j$ larger than the chosen threshold $n'$ shall be regrouped and the rest shall stay unchanged. While choosing the threshold $n'$ as the smallest class frequency, i.e. $n' = n_C$ is the natural choice, it maybe treated as a hyper-parameter and tuned for optimal performance in practice.

## 3 RELATED WORK

**Imbalanced learning using clustering**   Use of clustering methods in the context of imbalanced learning problems is not unknown. There are two families of resampling-based imbalanced learning methods using clustering: **1) Clustering-based upsampling**: K-means clustering has been used in conjunction with SMOTE upsampling method (Chawla et al., 2002) on the minorty class samples in order to avoid generating noisy samples when minority class distribution is multi-modal (Last et al., 2017). Clustering algorithms on the minority class samples have also been used to ensure a larger sampling frequency closer to the centroid (Singh & Dhall, 2018). Class-specific clustering has been used to ensure local class balance by enforcing a margin between the same and different class clusters (Huang et al., 2016; 2019); **2) Clustering-based downsampling**: Clustering has been used to improve upon the performance of random undersampling by first dividing the majority class

samples into clusters and randomly downsampling the samples within these clusters (Yen & Lee, 2009).

Closest to the proposed method is COG (Wu et al., 2010), which divides the large classes into multiple sub-classes by use of a clustering method. However, since their method was restricted to linear classifiers, the key motivation was to increase the complexity of the hypothesis class by allowing higher number of linear partitions in the low-dimensional data space. In contrast, we focus on image classification with DNNs, and our motivation stems from identifying underlying data distributions to help facilitate the representation learning.

**Pseudo labeling in semi-supervised learning**   Pseudo labeling is one of the most fundamental ideas used in semi-supervised learning setting, i.e. when small amount of labeled data accompanied with a large amount of unlabelled data is available at the training time. Self-training methods alternatingly perform two steps: 1) train the model on the labeled data and use it to assign pseudo labels to the unlabelled data, 2) retrain the model using labelled as well as a fraction of most confidently predicted unlabeled data along with their pseudo labels (Zhu, 2005). Check the recent surveys (Van Engelen & Hoos, 2020; Yang et al., 2021) for detailed literature review on pseudo labeling-based semi-supervised methods. In contrast to these methods, we fix the pseudo labeling at the start of the training process.

**Transfer learning for imbalanced data**   Transfer learning has also been explored in the imbalanced learning literature with the intention of: 1) transferring knowledge from head to tail class, 2) making use of unlabelled training data, 3) distill knowledge from a well-trained teacher model and, 4) finetuning the model on a more balanced training set; see (Zhang et al., 2021; Yang et al., 2022) for a detailed survey on these methods. Our proposed regrouping algorithm is most closely aligned with knowledge distillation, where we provide a new approach to perform transfer learning from a powerful CLIP model by imposing a structure on the representations of the majority classes.

**AP/AUPRC optimization**   The classical imbalanced learning techniques are aimed at optimizing the balanced accuracy. The idea of optimizing Area under Precison Recall Curve (AUPRC) / average precision (AP) directly has been extensively explored for classical ML methods, see (Qi et al., 2021) for literature survey. However, in the context of DNNs, they have been very briefly explored, mainly targeting image retrieval and object detection applications. FastAP (Cakir et al., 2019) and SmoothAP (Brown et al., 2020) have been proposed to handle the challenging AP objective by replacing it with a simpler approximation. RaMBO (Rolínek et al., 2020) and AP-Loss (Chen et al., 2019) use a blackbox differentiation and finite difference approximation methods respectively for optimizing AP. SOAP (Du et al., 2021) uses a surrogate loss to replace the AP objective and provide a novel stochastic algorithm to handle the composite structure in the objective with provable convergence guarantees. However, extending AP optimization to imbalanced multiclass problem has not been addressed in the literature. The natural extension of naively averaging the class wise AP metrics is misleading, since the AP metric is sensitive to imbalance ratio, and the effective imbalance ratio while calculating AP corresponding to each class varies in the case of imbalanced learning. Caliberating the precision values based on observation made in Williams (2021); Hoiem et al. (2012) before aggregating them may be a potential direction to explore in the future.

**Long Tailed Learning**   A subclass of IC problem recently emerging in computer vision is long-tailed classification (Zhang et al., 2021; Yang et al., 2022), which blends IC and few-shot learning as the tail classes are underrepresented in the training set. In comparison, in this paper, we assume that all classes are sufficiently represented despite the class imbalance. We observe from the comparison made in (Zhang et al., 2021) that classical IC methods fail to match the performance of state-of-the-art (SOTA) long tailed classification methods. However, we argue that further exploration of IC problems is still essential for two primary reasons: (1) numerous practical applications such as disease diagnosis, spam filtering, and fraud detection still fall under IC, and (2) SOTA long tailed methods often employ the use of standard IC-based methods within the overall framework, for e.g., Cao et al. (2019); Wang et al. (2020). In this paper, we focus on addressing the classical IC setting, and extension to long-tailed classification setting is left open for future work.

Table 1: **RG vs. others on binary CIFAR-100 and binary HAM10000 dataset**. ↑ means the larger, the better. The best scores are marked in **bold maroon**

| Method | binary CIFAR-100 | | | binary HAM10000 | | |
|---|---|---|---|---|---|---|
| | **BA** (%) ↑ | **AP** (%) ↑ | | **BA** (%) ↑ | **AP** (%) ↑ | |
| | | Neg $(49, 500)$ | Pos $(500)$ | | Neg $(9, 688)$ | Pos $(327)$ |
| CE | $81.0\pm_{2.00}$ | $99.9\pm_{0.00}$ | $64.8\pm_{1.90}$ | $66.8\pm_{1.40}$ | $98.7\pm_{0.30}$ | $48.8\pm_{4.30}$ |
| WCE | $87.3\pm_{1.40}$ | $99.9\pm_{0.00}$ | $58.5\pm_{0.90}$ | $79.4\pm_{1.80}$ | $99.7\pm_{0.00}$ | $45.7\pm_{7.60}$ |
| Focal | $77.2\pm_{1.20}$ | $99.9\pm_{0.00}$ | $64.7\pm_{1.10}$ | $65.6\pm_{3.30}$ | $99.0\pm_{0.20}$ | $49.3\pm_{0.80}$ |
| LDAM | $77.6\pm_{1.10}$ | $99.9\pm_{0.00}$ | $58.3\pm_{7.27}$ | $57.4\pm_{8.29}$ | $99.6\pm_{0.20}$ | $37.9\pm_{15.5}$ |
| LA | $79.3\pm_{1.30}$ | $99.9\pm_{0.00}$ | $65.3\pm_{1.00}$ | $64.4\pm_{2.60}$ | $98.9\pm_{0.10}$ | $47.0\pm_{1.00}$ |
| RUSC | $84.9\pm_{0.90}$ | $99.9\pm_{0.00}$ | $14.5\pm_{4.70}$ | $\textbf{85.9}\pm_{\textbf{0.70}}$ | $99.7\pm_{0.00}$ | $27.4\pm_{5.50}$ |
| DSMT | $79.2\pm_{0.70}$ | $99.9\pm_{0.00}$ | $58.7\pm_{11.0}$ | $66.9\pm_{1.10}$ | $99.2\pm_{0.10}$ | $52.9\pm_{3.50}$ |
| ROS | $81.2\pm_{1.70}$ | $99.8\pm_{0.00}$ | $68.5\pm_{2.10}$ | $66.4\pm_{1.50}$ | $99.7\pm_{0.00}$ | $54.4\pm_{2.00}$ |
| CLIP+MLP | $\textbf{91.9}\pm_{\textbf{0.70}}$ | $99.9\pm_{0.00}$ | $\textbf{84.2}\pm_{\textbf{1.00}}$ | $47.3\pm_{1.90}$ | $98.6\pm_{0.00}$ | $0.70\pm_{0.00}$ |
| RG | $84.5\pm_{2.20}$ | $99.7\pm_{0.00}$ | $73.8\pm_{1.10}$ | $71.3\pm_{3.10}$ | $98.8\pm_{0.15}$ | $55.5\pm_{3.77}$ |
| RG-WCE | $85.1\pm_{1.00}$ | $99.8\pm_{0.00}$ | $72.7\pm_{1.90}$ | $75.1\pm_{1.60}$ | $98.9\pm_{0.20}$ | $\textbf{63.8}\pm_{\textbf{1.90}}$ |

## 4 EXPERIMENTS

### 4.1 SETTING

In this section, we compare our proposed regrouping (RG) method with standard ERM based on cross-entropy loss (CE) and several other state-of-the-art methods from two major categories: 1) loss-based methods such as Weighted cross-entropy loss (WCE), Focal loss (Lin et al., 2017), LDAM (Cao et al., 2019), Logit Adjustment (LA) (Menon et al., 2020), AP-Loss (Chen et al., 2019), and 2) resampling-based methods such as Random under-sampling by cluster (RUSC) (Yen & Lee, 2009), Deep SMOTE (DSMT) (Dablain et al., 2022), and Random over-sampling (ROS). We apply grid search for methods involving tunable hyper-parameters and report the best performance. In RG, we use CLIP as the image encoder, k-means for the clustering algorithm, cross-entropy loss as the loss function, and max as the aggregation method. We repeat the experiments with 3 runs and reported the means and standard deviations. Details of the experiment setups can be found in Appendix A, ablation study to explore the impact of the clustering algorithm, image encoder, loss function, and aggregation method can be found in Appendix B, and exploration of RG in sentiment classification with language models can be found in Appendix C.

**Dataset** We study three datasets, including: **1) Binary CIFAR-100**. The original CIFAR-100 is a balanced dataset with $50000$ images of $100$ classes. We modify it by taking out the worm images as the positive class and combining the rest as the negative class, resulting in an imbalanced binary dataset with an imbalance ratio of $99 : 1$. **2) Binary HAM10000** HAM10000 (Tschandl et al., 2018) is a large collection of multi-source dermatoscopic images of 7 common skin lesions: Melanocytic nevi (nv), Melanoma (mel), Benign keratosis-like lesions (bkl), Basal cell carcinoma (bcc), Actinic keratoses (akiec), Vascular lesions (vasc), and Dermatofibroma (df). Similar to binary CIFAR-100, we binarize it by grouping 6 classes (nv, mel, bkl, akiec, vasc, and df) as a negative class, and leaving 1 class (bcc) as the positive class. **3) HAM10000** We train and evaluate the original imbalanced multiclass classification task.

**Evaluation Metrics** We evaluate all the methods using both BA and average precision (AP). Our consideration of including AP is that BA is highly specialized for recall and ignores precision which can be biased and misleading, as discussed in Sec. 1. To ensure fair comparisons, metrics such as AUPRC (i.e. area under the precision and recall curve) and $F_1$ score (i.e., the harmonic mean of precision and recall) that consider both precision and recall are more appropriate. However, $F_1$ score depends on the decision rule, specifically, how continuous-valued prediction scores are discretized into discrete class labels. It is known that the typical choice of $0.5$ threshold, used in balanced scenarios, is provably suboptimal in the presence of class imbalance (Menon et al., 2013; Singh & Khim, 2022). Considering that AUPRC is agnostic to the decision rule and is often approximated

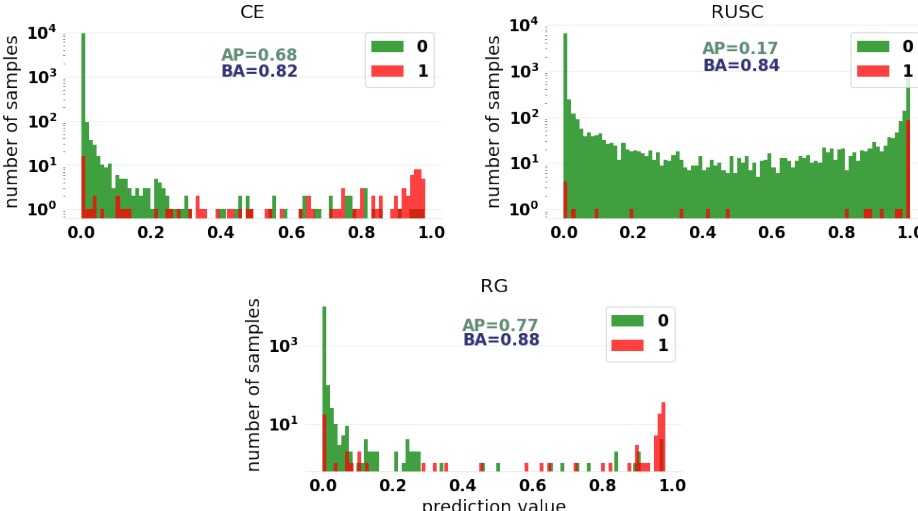

Figure 7: prediciton distribution of CE, RUSC, and RG (ours) on CIFAR100 dataset. BA is a biased metric: while RUSC achieves a higher BA score than CE, it suffers a lot from its high false-positive rate. Our proposed RG methods achieve the highest AP score. By default, we set the decision threshold as 0.5 for all methods.

by AP given finite sample sizes (Boyd et al., 2013), we report AP as the primary measure of IC performance for all subsequent experiments.

## 4.2 BINARY CLASSIFICATION

We start with binary IC problems on two datasets – binary CIFAR-100 and binary HAM10000. In Tab. 1, we observe that BA and AP often give divergent results. For example, on two datasets, RUSC is performing very competitively in terms of BA but worse in AP. As illustrated ahead, we believe it is due to the inferiority of BA when evaluating highly imbalanced datasets. To confirm our hypothesis, we plot the distribution of the prediction score and compare it with three representative methods in Fig. 7. It is clear that RUSC performs the worst as it struggles to separate between positives and negatives while the BA is still very high, which indicates that BA can give very misleading results when evaluating a highly imbalanced dataset. Notably, BA is sensitive to the choice of the decision threshold. For CE and RG, one may need to carefully tweak the threshold to get a high BA. In contrast, RG achieves much better distribution and is more lenient on the threshold choice. As CLIP is introduced in the image encoding process of RG, it is worth exploring if the CLIP model representations are sufficient to train a classifier. Therefore, we set an extra competitor which takes CLIP as a fixed feature extractor, and train a multi-layer perceptron (MLP) as the classifier. In CIFAR-100, CLIP does help to improve the performance and outperform RG. However, we observe a significant performance drop when switching to HAM10000, a more challenging dataset with higher image resolution from a specialized domain – different from where CLIP was pretrained. We believe that instead of direct feature transfer, RG helps to find better label structures, which further guide the model to learn good feature representations and thus improve the performance.

## 4.3 MULTICLASS CLASSIFICATION

Now we turn to a real-world medical multiclass classification task on the HAM10000 dataset. Unlike the binary setting, where we can clearly tell the minority and majority class, multiclass classification can be more complex as it may have multiple small/large classes. A natural extension of our method would be to set a hard threshold $n'$, and for any class with samples below the bar, we treat it as a minority class and keep the original labels. We regroup the data using a clustering algorithm for those large classes with sufficient samples. If the data acquisition process is known, the number of

Table 2: **RG vs. others on HAM10000 dataset**. We report both BA and class-wise AP scores. Below the class name is the corresponding number of images in each class. ↑ means the larger, the better. The best scores (with interval intersects) are marked in **bold maroon**

| Method | BA (%) ↑ | AP (%) ↑ | | | | | | |
|---|---|---|---|---|---|---|---|---|
| | | nv 6705 | mel 1113 | bkl 1099 | bcc 514 | akiec 327 | vasc 142 | df 115 |
| CE | $55.4_{\pm 2.20}$ | $96.2_{\pm 0.40}$ | $59.2_{\pm 1.50}$ | $65.5_{\pm 2.60}$ | $73.2_{\pm 2.70}$ | $54.0_{\pm 1.60}$ | $76.0_{\pm 4.50}$ | **$47.5_{\pm 5.40}$** |
| WCE | **$64.2_{\pm 1.20}$** | $96.1_{\pm 0.10}$ | $46.3_{\pm 3.30}$ | $54.3_{\pm 3.00}$ | $62.9_{\pm 2.70}$ | $53.5_{\pm 6.60}$ | **$83.1_{\pm 2.00}$** | **$51.0_{\pm 4.30}$** |
| Focal | $56.7_{\pm 0.70}$ | $96.5_{\pm 0.30}$ | $59.6_{\pm 1.60}$ | $66.0_{\pm 0.80}$ | $72.6_{\pm 1.30}$ | $54.6_{\pm 1.80}$ | $80.6_{\pm 1.80}$ | $47.7_{\pm 5.10}$ |
| LDAM | $42.1_{\pm 3.40}$ | $94.2_{\pm 0.30}$ | $34.5_{\pm 1.90}$ | $37.7_{\pm 2.10}$ | $44.4_{\pm 11.3}$ | $34.3_{\pm 2.40}$ | $61.0_{\pm 10.5}$ | $7.50_{\pm 2.10}$ |
| LA | $56.2_{\pm 1.30}$ | $96.8_{\pm 0.10}$ | $59.9_{\pm 0.40}$ | $64.1_{\pm 2.20}$ | $73.4_{\pm 4.60}$ | $51.5_{\pm 3.20}$ | **$81.4_{\pm 4.00}$** | $42.5_{\pm 3.50}$ |
| RUSC | $50.1_{\pm 3.60}$ | $90.7_{\pm 0.30}$ | $25.7_{\pm 2.10}$ | $28.6_{\pm 2.40}$ | $28.3_{\pm 5.60}$ | $28.5_{\pm 1.40}$ | $61.7_{\pm 7.00}$ | $17.3_{\pm 6.20}$ |
| DSMT | $56.2_{\pm 2.40}$ | $96.6_{\pm 0.30}$ | $61.3_{\pm 0.20}$ | $65.9_{\pm 3.60}$ | $74.0_{\pm 2.10}$ | $54.7_{\pm 2.40}$ | $73.3_{\pm 4.90}$ | $41.4_{\pm 1.60}$ |
| ROS | **$66.8_{\pm 2.10}$** | **$97.9_{\pm 0.10}$** | **$72.7_{\pm 1.00}$** | **$81.8_{\pm 1.30}$** | **$84.8_{\pm 1.40}$** | **$70.2_{\pm 3.70}$** | $85.5_{\pm 4.20}$ | $47.0_{\pm 2.70}$ |
| CLIP+MLP | $22.5_{\pm 0.40}$ | $90.5_{\pm 0.80}$ | $41.2_{\pm 0.50}$ | $22.5_{\pm 1.20}$ | $11.0_{\pm 1.60}$ | $6.50_{\pm 1.80}$ | $2.00_{\pm 0.60}$ | $1.50_{\pm 0.00}$ |
| RG | **$62.6_{\pm 2.20}$** | $93.7_{\pm 0.60}$ | $65.0_{\pm 1.70}$ | $72.1_{\pm 1.20}$ | $75.8_{\pm 3.30}$ | $57.4_{\pm 5.80}$ | **$80.8_{\pm 2.60}$** | **$54.5_{\pm 4.10}$** |
| RG+WCE | **$63.5_{\pm 1.60}$** | $91.7_{\pm 0.20}$ | $60.4_{\pm 1.80}$ | $67.6_{\pm 2.50}$ | $72.9_{\pm 2.10}$ | $54.7_{\pm 5.30}$ | **$81.1_{\pm 2.20}$** | **$54.7_{\pm 1.90}$** |

clusters can be determined by the intrinsic structure of the data. If no prior knowledge is provided, as is in this task, a natural choice is to set the number of clusters as $\frac{n_i}{n}$ where $n_i$ is the number of images in the current class. We exclude AP in this task as it performs poorly in binary classification, and is sensitive to high imbalance ratios.

As shown in Tab. 2, our method significantly outperforms all loss-based methods (e.g., CE, WCE, Focal, LDAM, and LA) without introducing extra data during training. Surprisingly, ROS also outperforms many loss-based methods, which is likely due to the highly imbalanced training data. For highly imbalanced datasets, given the limited batch size and without augmenting minority samples, many batches will have insufficient or no samples for minority classes, making gradients less informative. Despite the high performance, ROS adds more computation cost than other methods as it introduces $4\times$ training data. Notably, our methods only add tolerable computation cost in the regrouping phase while achieving the best AP scores for the minority class with minimal performance loss in the majority classes.

## 5 CONCLUSIONS

In this paper, we propose a novel simple yet effective baseline for imbalanced classification called **R**e**G**rouping (RG), which involves decomposing the majority class to form smaller clusters and thus induces a more balanced dataset that adheres to the latent label hierarchy. We first use a motivating example to show that the proposed method is able to learn efficient representations for image data and synchronize the learning pace for different classes as well. Then, we compare RG with several popular imbalanced learning methods and show that RG is a strong competitor among all methods and often gives the best performance. Lastly, we shed light on the ineffectiveness of commonly used performance metrics such as BA in terms of comparing different methods for imbalanced classification and recommend the use of AP/AUPRC metrics for this purpose.

Although we mostly discuss imbalanced classification with image data, we believe our method can be readily extended to text data with popular foundation language models as text encoders. As a next step, our future work will focus on: 1) extending RG to other data modalities; 2) avoiding the dependency on pretrained encoders by using self-supervised learning techniques, and 3) studying more complex and challenging imbalanced tasks such as long-tail learning. We believe that our methods give a fresh perspective to the imbalance learning problem which gives new opportunities and challenges for future research.

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
