## A    EXPEIMENT DETAILS

**Implementation Details**    For HAM10000 dataset, we resize the images into $224 \times 224$ and normalize them by the ImageNet mean and standard deviation. For CIFAR-100 dataset, we resize the image into $32 \times 32$ and follow the same normalization pipeline. We apply mild random rotation and horizontal flips for augmentation. For training, we use ResNet34 and SGD with a batch size of 256. We initialize the learning rate as 0.01 and adjust it using the CosineAnnealingLR scheduler during training. We train the model for 400 epochs and evaluate the final model by BA and class-wise AP. For the RG model, we first map the images into a low-dimension feature space using the CLIP model as a feature extractor. Then we apply K-means on the feature space for regrouping where images in the same group are assigned with a new pseudo-label. The number of clusters is determined by the ratio of the current class to the smallest class. For models with hyperparameters, we adopt grid-search and specify a wide range for searching. Details about hyperparameter tuning can be found in supplementary materials. All the experiments are conducted in an environment consisting of AMD 7763 processer and NVIDIA A100 GPU.

**Hyprameter Tunning**    To ensure a fair comparison, we apply a grid search for each method to find the best hyperparameter and pick the best model.

- Focal: Focal loss has a hyperparameter $\gamma$. We construct the search scope as $\gamma \in \{0.5, 1, 1.5, 2, 2.5, 3\}$

- LDAM: LDAM has two hyperparemter $max_m$, and $m$. We define the search grid as $max_m \in \{0, 0.5, 1\}$ and $s \in \{10, 20, 30\}$

- LA: LA has one hyperparameter $\tau$. We define the search scope as $\tau \in \{0, 0.5, 1, 1.5, 2, 2.5\}$.

- RG: RG model has a hyperparameter of $threshold$ which indicates the minimum samples of a class that is not to be regrouped.

  In binary CIFAR-100 datasets, we define the search grid as $threshold \in \{300, 400, 500, 600, 700, 800\}$.

  In binary HAM10000, we defined it as $threshold \in \{250, 300, 350, 400, 450, 500\}$

  In HAM10000, we defined it as $threshold \in \{200, 300, 400, 500, 600, 700\}$.

## B    ABLATION STUDY

**Impact of regrouping methods**    We study the impact of different regrouping approaches on classification performance. We compare three methods including: 1) **Random:** random regroup the majority class by uniformly assigning the pseudo labels; 2) **RP+K-means:** randomly project the image into a low-dimensional feature space using an untrained neural network, and regroup using K-means; and 3) **CLIP+K-means:** use a pretrained model (CLIP) as the backbone for feature extraction and then apply K-means. We conducted the experiments on a binary CIFAR100 dataset and calculated the AP and BA scores; see Fig. 8 (b) for a summary of results. The results demonstrate that grouping quality is crucial for the final classification performance. Using a model pretrained on a large-scale dataset (CLIP) as a backbone feature extractor for clustering can be good practice. Methods that generate a balanced dataset but randomly group the data do not necessarily improve the performance, implying that the regrouping must act in coordination with the intrinsic label structure.

**Impact of loss functions**    To study whether RG is sensitive to loss function, we compared the performance of RG with different losses including: CE, WCE, LA, LDAM, and Focal. In Fig. 8 (c), we show that RG uniformly improves performance regardless of which metric to evaluate and is not sensitive to the choice of loss functions.

**Logit aggregation: sum or max?**    In practice, we can aggregate the logit of the negative outputs by taking either the max or sum. In Fig. 8 (d), we compare the two aggregation methods. We see that max aggregation performs better, but overall it is negligible. Thus, we suggest trying both whenever possible.

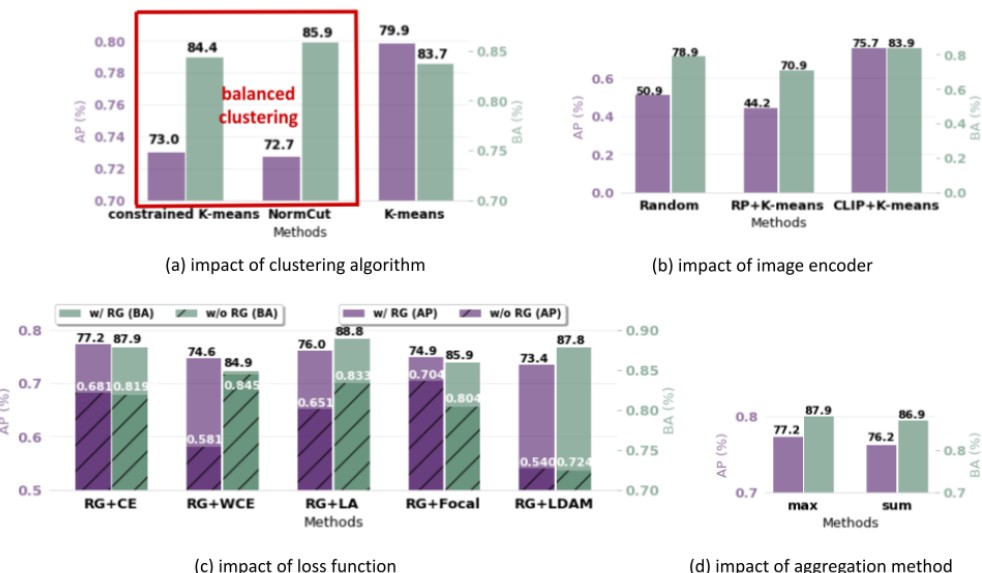

Figure 8: Ablation studies.

**Impact of the clustering algorithm**  The size of the cluster sometimes varies. To study the impact of the imbalance among clusters, we select two balanced clustering algorithms: constrained K-means (Bradley et al., 2000) and normalized cut (Shi & Malik, 2000), and compare them with the standard K-means. We find that purposely making it strictly balanced does not always guarantee better performance. Without bells and whistles, naive K-means lead to better AP scores and are therefore recommended for practical use.

## C  EXTENDED EXPERIMENTS

In this section, we provide an extended experiment on a language task to show the promise of applying our methods beyond image classification. More specifically, we take the huggingface financial phasebank dataset (Malo et al., 2014), which consists of 4840 sentences of English-language financial news categorized by sentiment with a label frequency of $2,879$ (neutral), $1,363$ (positive) and $604$ (negative). We remove DSMT in the comparision as it is not straightforward to generate synthetic data using pairs of texts. We also remove RUSC because it gives inferior performance, as demonstrated in 7. For RG implementation, we take Bert-base-uncased model as the text encoder to generate features for K-mean to assign pseudo-labels. For sentiment classificaiton, all the methods take the same Bert-base-uncased model and fine-tune from pre-trained weights.

Table 3: **RG vs. others on financial phasebank dataset**. Below the class name is the corresponding number of images in each class. ↑ means the larger, the better. The best scores (with interval intersects) are marked in **bold maroon**

| Method | BA (%) ↑ | AP (%) ↑ | | |
| --- | --- | --- | --- | --- |
| | | Neutral (2,879) | Positive (1,363) | Negative (604) |
| CE | $84.2_{\pm0.61}$ | $94.5_{\pm1.76}$ | $\mathbf{88.4_{\pm2.68}}$ | $89.9_{\pm1.64}$ |
| WCE | $83.7_{\pm0.64}$ | $\mathbf{94.5_{\pm1.46}}$ | $86.5_{\pm2.71}$ | $90.5_{\pm2.93}$ |
| Focal | $84.9_{\pm0.90}$ | $93.7_{\pm2.12}$ | $85.9_{\pm4.69}$ | $90.5_{\pm1.63}$ |
| LDAM | $83.8_{\pm0.89}$ | $94.1_{\pm1.58}$ | $87.4_{\pm2.36}$ | $90.4_{\pm2.15}$ |
| LA | $84.3_{\pm0.59}$ | $94.1_{\pm1.98}$ | $87.1_{\pm2.18}$ | $90.9_{\pm0.57}$ |
| ROS | $84.0_{\pm0.64}$ | $94.0_{\pm1.22}$ | $86.8_{\pm2.98}$ | $90.6_{\pm2.52}$ |
| RG | $86.2_{\pm0.45}$ | $93.5_{\pm0.53}$ | $88.1_{\pm1.43}$ | $91.9_{\pm1.30}$ |
| RG+WCE | $\mathbf{87.5_{\pm0.35}}$ | $94.3_{\pm0.85}$ | $88.7_{\pm1.01}$ | $\mathbf{93.6_{\pm0.53}}$ |

We can see from Tab. 3 that RG, especially when combined with WCE, greatly outperforms other baseline methods. We hypothesize that RG enables the model to find more fine-grained label structures for the "neutral" class, thus introducing more rich information for the classifier to better comprehend the text sentiment.