# OpenReview forum: "Mining latent labels for imbalance classification: a regrouping perspective"
_ICLR.cc/2024/Conference — Submitted to ICLR 2024_

### Official Review · Reviewer_9T2x · 2023-10-20

**Soundness:** 2 fair
**Presentation:** 2 fair
**Contribution:** 1 poor
**Rating:** 3
**Confidence:** 4

**Summary:**

To handle imbalanced data, the authors propose RG (Regrouping) by
clustering instances in the majority class, create pseudo-classes
from the classes, and learned a classifier with more classes.  If the
score for the minority class is larger than the pseudo-class, the
minority class is predicted; otherwise, the majority class is
predicted.  For multi-class data, all classes except the smallest
class are regrouped.  The proposed approach is quite straightforward.

For binary classification, RG was evaluated on two datasets.  For
multi-class classification, RG was evaluated on one dataset.

**Strengths:**

The problem of imbalanced data in multi-class classification is
interesting.  RG outperforms existing techniques in balanced accuracy
(BA) in one dataset.

**Weaknesses:**

The efficacy of RG is not well demonstrated.  In Table 1, RG only
outperforms in one of 3 metrics in one of 2 datasets.  For the
multi-class problem only one dataset is used.

The choice of datasets could be improved.  The 9 classes in CIFAR 10
are quite different and merging to simulate a single majority class
might have a quite diversified class.  For example, a majority class
has many images of dogs, but the different kinds (subclasses/clusters)
of dogs have commonalities to be dogs.  Hence, Binary CIFAR 10 might
not be a good dataset to use.  Binary HAM10000 on "dermatoscopic
images of 7 common skin lesions" is more appropriate.

The presentation could be improved.  For example:

Sec 2.2: Sum aggregation was discussed, but it seems to be not used
in any experiments.  Also, the motivation for Sum aggregation was not
discussed.

Sec. 2.2 does not discuss how clusters are formed via regrouping.
k-means is mentioned in 4.1 Setting of experiment.

**Questions:**

RG+WCE: since RG tries to balanced class sizes, why do you need WCE
(weighted cross entropy)?  Why did WCE help?  Could you describe WCE
or cite a source?

p7.  "Considering that AUPRC is agnostic to the decision rule and is
often approximated by AP given finite sample sizes"--any citations or
evidence to support the statement?

Fig. 5, caption: airplane, not apple ?

---

> ### Author Response · Authors · 2023-11-16
> **Authors' Response**
>
> We would like to heartfully thank the reviewer for their considerate comments and critiques of our work. In the following, we have addressed some of the questions raised in the review. Please feel free to follow up if any aspects are unclear, or if there are further questions.
>
> - More datasets: We acknowledge the limitation that we only have one multi-class dataset, and have addressed this issue by extending our experiment to include a new multi-class task. Specifically, we take the financial phasebank dataset (Malo et al. (2014)), which consists of 4840 sentences of English-language financial news categorized by sentiment with a label frequency of 2,879 (neutral), 1,363 (positive), and 604 (negative). We have shown that in the new dataset, RG surpasses other baselines and remains dominant. For more information about the experiment setups and results, we would kindly refer to Appendix C for details.
> - Choice of dataset for visualization experiments: We appreciate the reviewer’s comment on how binary HAM10000 can form a more realistic dataset for visualization experiments. The choice of the CIFAR-10 dataset for the visualization experiment is based on three arguments:
>     - Popularity: CIFAR-10 is one of the most popular datasets in image classification, and the class labels are easily interpretable by the readers, and hence a natural choice for a motivating example
>     - Relevance to the regrouping idea: The fact that the majority class is diverse makes it an ideal situation for regrouping. Hence, it helps in highlighting the failure of naively using clustering algorithms to separate samples of different classes, for example in raw image space and autoencoder latent representation space.
>     - Difficulty:  As seen from the results in Section 4, the classification of HAM10000 images is a much harder task in comparison to CIFAR10 image classification, and hence presents a more clear motivation for the readers.
>
>     While a dataset like CATSvsDOGS could potentially be a more realistic setting for imbalance classification, the lack of fine-grained labels (for example: breeds of the animals) makes it difficult to visualize the effect of regrouping. Additionally, the discrepancy between how DNNs interpret label structures and human understanding further complicates the analysis.
> - Presentation: We would like to thank the reviewer for the comments on the presentation, we have addressed them as follows:
>     - Aggregation: We performed an ablation study to show the impact aggregation method in Appendix B, where max aggregation is found to be marginally better in comparison to sum aggregation, but since the difference is not too large it is recommended to try both methods. We have added some explanation for both aggregation methods in Sec 2.2 to avoid confusion.
>     - Clustering Method: In Appendix B, we also explored the role of clustering algorithms in the performance of the proposed method, especially focussing on whether enforcing equal cluster sizes can be beneficial to the method. Two balanced clustering algorithms: constrained K-means (Bradley et al. (2000)) and Normalized Cut (Shi et al. (2000)) have been compared with the naive K-means method. We find that balanced clustering can lead to slightly better BA, but perform much worse in terms of AP metric, and hence the choice of a simple K-means algorithm is proposed. To avoid confusion, we have also added brief information about the clustering algorithm in Sec 2.2.
>     - Image Caption: We have fixed the typo in the image caption, it should have been airplane instead of apple.
> - Why WCE is required with RG?: WCE is the weighted version of cross entropy loss where each sample is weighted by the inverse of the corresponding class frequency, so that classes with fewer samples get higher weights, check Elkan (2001) for the general concept of reweighting. Since we are using K-means algorithm to perform clustering and assign pseudo labels, the resulting classes are not guaranteed to be balanced. Hence, along with CE, we have also reported results that use WCE loss function.
> - AUPRC/AP metric clarification: AUPRC is a metric that calculates the area under precision-recall curve, where each point is based on a specific threshold (decision rule). Hence, AUPRC (similar to AUROC) is independent of the choice of decision rule. Average Precision (AP) is known to be an unbiased estimate of AUPRC (Boyd et al. (2013)). We have added the reference in the paper as well.

---

> > ### Author Response · Authors · 2023-11-16
> > **Authors' Response (cont.)**
> >
> > References:
> > - Elkan, C. (2001, August). The foundations of cost-sensitive learning. In International joint conference on artificial intelligence (Vol. 17, No. 1, pp. 973-978). Lawrence Erlbaum Associates Ltd.
> > - Boyd, K., Eng, K. H., & Page, C. D. (2013). Area under the precision-recall curve: point estimates and confidence intervals. In Machine Learning and Knowledge Discovery in Databases: European Conference, ECML PKDD 2013, Prague, Czech Republic, September 23-27, 2013, Proceedings, Part III 13 (pp. 451-466). Springer Berlin Heidelberg.
> > - Bradley, P. S., Bennett, K. P., & Demiriz, A. (2000). Constrained k-means clustering. Microsoft Research, Redmond, 20(0), 0.
> > - Shi, J., & Malik, J. (2000). Normalized cuts and image segmentation. IEEE Transactions on pattern analysis and machine intelligence, 22(8), 888-905.
> > - P. Malo, A. Sinha, P. Korhonen, J. Wallenius, and P. Takala. Good debt or bad debt: Detecting semantic orientations in economic texts. Journal of the Association for Information Science and Technology, 65, 2014.

---

> > ### Comment · Reviewer_9T2x · 2023-11-21
> > **comments on response**
> >
> > Thanks for the response.
> >
> > I don't think having subclasses in the majority class is important because a majority class might not have (defined) subclasses--the majority class might just has a lot of instances.  Also, if there are (defined) subclasses, one might evaluate how well the clustering algorithm recovers the subclasses.  However, the clustering algorithm is the well-known k-means algorithm, not one that is newly proposed.
> >
> > What is the motivation for Sum aggregation?

---

### Official Review · Reviewer_AJcJ · 2023-10-28

**Soundness:** 2 fair
**Presentation:** 2 fair
**Contribution:** 2 fair
**Rating:** 3
**Confidence:** 5

**Summary:**

This paper proposes a regrouping method to improve the performance of imbalanced learning, which decomposes the majority classes into subclasses by clustering and trains the model under the extended classes. The authors analyzed the ability of the proposed RG , demonstrating its ability to facilitate learning efficient representation and synchronizing the training progress across different classes, and verified the performance thorough a range of experiments.

**Strengths:**

1) ReGrouping method is different from the conventional loss reweighting or re-sampling methods that changes the class importance explicitly. By regrouping, the learning pace of each class (especially for rare classes) can be directly intervened as shown in the loss variation illustration in Figure 5.

2) The authors provided some interesting validations to support the design about the proposed method like the synchronous learning, and presented how to design the clustering number and extend to multi-class learning as well as the underlying tricky points for the optimal performance.

3) The authors conducted a range of experiments on both binary and multi-class imbalanced learning tasks, demonstrating superior performance compared to state-of-the-art methods in terms of balanced accuracy (BA) and average precision (AP) metrics.

**Weaknesses:**

Although the methods shows the interesting points of the proposed regrouping method, some critical concerns remained and are summarized as follows.

1) The novelty concern can be a big problem. As the authors mentioned about the COG method (local clustering for imbalanced learning in Wu, et. al., 2010), both the proposed method and COG shares the same spirit for imbalanced learning, and the technical major difference is COG follows the SVM classifier. Despite in different data context, they are both for imbalanced learning, which weakens the novelty of this work.

2) The technical description is not sufficient, as we can see that there is lack of the clustering ways for pseudo labels that are used in the regrouping method. This also connects the lack of the corresponding experiments to verify the clustering impact on the final performance. Especially, as shown in Figure 5 and Figure 6, how to assign the pseudo labels does matter about the performance, which makes the readers care about the clustering effectiveness.

3) The experiments are also not very persuasive although some experiments have shown the improvement about RG. The major concern is about the datasets and the baselines especially for the multi-class classification experiments. There are a range of explorations in long-tailed learning for multi-class classification problems. However, we cannot find any sufficient comparison with the recent advances like Decoupling, LA (logit adjustment), BCL and so on. For the datasets in long-tailed learning, CIFAR100-LT, ImageNet-LT or INaturalist are all widely adopted benchmarks, which should be included in this submission.

**Questions:**

Overall, I am interested in this regrouping idea for imbalance learning, although it has been proposed in previous explorations. What is the intrinsic difference for imbalanced learning should be highlighted, instead of some minor difference as in the description of the submission. For other questions, please see above weakness.

---

> ### Author Response · Authors · 2023-11-16
> **Authors' Response**
>
> We would like to heartfully thank the reviewer for their considerate comments and critiques of our work. In the following, we have addressed some of the questions raised in the review. Please feel free to follow up if any aspects are unclear, or if there are further questions.
>
> - Novelty: We agree with the reviewer that the proposed method shares a high-level overview with Classification using local clustering (COG), proposed by Wu, et. al. (2010) to tackle imbalanced classification. However, we would like to highlight some key arguments in the paper that differ from COG:
>     - Motivation: COG was proposed in the context of linear SVM models in order to: a) decompose the majority class samples so that they become linearly separable from minority class samples, and b) produce sub-classes with relatively uniform sizes. Note that a) is irrelevant in the context of deep neural network (DNN) models, and b) is not supported (in Wu, et. al. (2010)) with any theoretical/empirical evidence. In contrast, our motivation for this method stems from the advantages demonstrated through the use of multiclass labels, for the one-vs-rest CIFAR-10 classification problem, over binary labels.
>     - Insights: Wu, et. al. (2010) justify their method by showing how linear decision boundaries between pseudo-classes found by COG can separate linearly inseparable classes. However, for DNN classifiers it is known that even random labels can fit perfectly on a training set (Zhang et al. (2021)), and hence the insight from Wu et al. (2010) is not valid for DNNs. In comparison, in this paper, we show the ability of the proposed method to learn meaningful representations (Figure 3 bottom pane), synchronize training across different labels (Figure 5), and achieve better performance (Figure 6). These insights shed light on the training dynamics in binary vs multiclass setting for the first time to the best of our knowledge, which could also be of independent interest.
>     - Implementation: In COG, pseudo labels are assigned by clustering the data in data space, we propose to cluster in the image encoding space. The choice of image encoder is central to the merits of the proposed method – it can be seen in Figure 3 (top pane) that naively clustering the data in raw image space, or even the representation space of autoencoder, will lead to very noisy pseudo labels and would result in poor final performance. We argue that this implementation detail of the proposed method is non-trivial for a practitioner implementing COG method for DNNs.
>
> 	To conclude, we argue that this work provides a fresh perspective for exploring and expanding on the method proposed by Wu et al. (2010) as a powerful baseline method to tackle imbalance.
> - Clustering Method: We would like to thank the reviewer for pointing out the lack of clarity about the clustering algorithm in the method description, we have updated the paper to make it more clear. Essentially, the proposed method is agnostic to the choice of clustering algorithm, but the conventional K-means algorithm seems to perform well empirically. Note that the proposed method depends heavily on the choice of image encoder (which has been explored in detail in Section 2.1), but less so on the choice of clustering algorithm used in the image encoder space. A natural question could be if enforcing the clustering algorithm to generate relatively balanced clusters could be helpful, which is explored in the ablation study in Appendix B. Two balanced clustering algorithms: constrained K-means(Bradley et al. (2000)) and Normalized Cut (Shi et al. (2000)) have been compared with the naive K-means method. We find that balanced clustering can lead to slightly better BA, but perform much worse in terms of AP metric, and hence the choice of a simple K-means algorithm is proposed.
> - Comparison for long-tailed learning dataset: Extension of the proposed method to long-tailed learning setting has been left open for future work. This requires careful handling of the tail classes, and addressing the explosion in numbers of pseudo-classes due to the extremely small sample size in tail classes. We have briefly discussed the merits of exploring classical imbalance classification methods in the last paragraph of Section 3: numerous practical applications still fall under classical imbalance setting, and classical methods often form a base for long-tailed learning methods. The competing methods have also been chosen accordingly. To show that our proposed method can work well in a classical imbalance classification setting, we added an extended experiment using the financial phasebank dataset (Malo et al. (2014)). The details of the experiment setup and results can be found in the revised appendix. For a quick summary, we observed consistent improvements of RG over other baselines, both in terms of BA and AP.

---

> > ### Author Response · Authors · 2023-11-16
> > **Authors' Response (cont.)**
> >
> > References:
> > - Wu, J., Xiong, H., & Chen, J. (2010). COG: local decomposition for rare class analysis. Data Mining and Knowledge Discovery, 20, 191-220.
> > - Bradley, P. S., Bennett, K. P., & Demiriz, A. (2000). Constrained k-means clustering. Microsoft Research, Redmond, 20(0), 0.
> > - Shi, J., & Malik, J. (2000). Normalized cuts and image segmentation. IEEE Transactions on pattern analysis and machine intelligence, 22(8), 888-905.
> > - P. Malo, A. Sinha, P. Korhonen, J. Wallenius, and P. Takala. Good debt or bad debt: Detecting semantic orientations in economic texts. Journal of the Association for Information Science and Technology, 65, 2014.

---

> ### Comment · Reviewer_AJcJ · 2023-11-22
>
> Thanks for the response. After reading the rebuttal, I appreciate the authors' effort and encourage the authors to further refine their works. The rating is remained given the current unsolved concerns.
>
> Best,
>
> The reviewer.

---

### Official Review · Reviewer_nZ1i · 2023-10-31

**Soundness:** 3 good
**Presentation:** 3 good
**Contribution:** 2 fair
**Rating:** 5
**Confidence:** 2

**Summary:**

This paper proposes a simple solution for class imbalanced problem by grouping the majority class to smaller sub-classes. The paper is well-written and easy to read.

**Strengths:**

Showcasing how multi-class classification by regrouping the majority class to smaller sub-classes work better than a binary classification.

**Weaknesses:**

- In the experiments there are no error bars.
- There is no experiment that any model that is not data hungry has been applied to compare it with DNN.

**Questions:**

- I would like to see the experiments results with error bars included. For example if you run the experiment n times and calculate standard deviation.
- I would also like to see how the results change if you apply non-hungry methods such as Gaussian processes.
- Sometimes groping the classes to small sub-groups is a difficult task by itself, how do you decide what type of data you can use to have this meaningful sub-groups? What happens if you can't put them into smaller groups?
- What are the limitations of your method?

---

> ### Author Response · Authors · 2023-11-16
> **Authors' Response**
>
> We would like to heartfully thank the reviewer for their considerate comments and critiques of our work. In the following, we have addressed some of the questions raised in the review. Please feel free to follow up if any aspects are unclear, or if there are further questions.
>
> - Error Bars: We have rerun all the experiments with 3 random seeds and report the mean with standard deviation, please see the updated results in the revised manuscript.
> - Gaussian Processes: In this paper, we mainly focus on classification based on deep learning models, especially in the image domain. We could find only two works that address Gaussian Process-based classification for image data (Blomqvist et al. (2020), Van der Wilk et al. (2017)), but their public code repositories have not been maintained and we are unable to implement these methods from scratch due to limited time of the discussion period. However, the method proposed by Wu et al. (2010) is similar to the proposed method for tabular data where results on linear SVM models are provided, which may be of interest to the reviewer.
> - Meaningful subgroups: A potential way to check whether the data can be partitioned into sub-groups is to make use of a visualization tool such as t-SNE plot (For example: As used in Figure 3 top pane). When data cannot be put into meaningful subgroups, it might point to the choice of encoder model (i.e. CLIP model in our case), and other encoder models can be tried.
> - Limitations: The key limitation of the proposed method is to handle long-tailed datasets, where the tail classes lie in the few-shot learning setting (i.e. tail classes have few training samples). This leads to an explosion in the number of pseudo-classes, and hence extension to this setting is non-trivial and left open for future work. Another limitation is that the number of pseudo-classes (K) in the current format is a tuning hyper-parameter, a future work could try to deduce this in a more methodological manner to save excess computation for tuning the hyper-parameter.
>
> References:
> - Blomqvist, K., Kaski, S., & Heinonen, M. (2020). Deep convolutional Gaussian processes. In Machine Learning and Knowledge Discovery in Databases: European Conference, ECML PKDD 2019, Würzburg, Germany, September 16–20, 2019, Proceedings, Part II (pp. 582-597). Springer International Publishing.
> - Van der Wilk, M., Rasmussen, C. E., & Hensman, J. (2017). Convolutional gaussian processes. Advances in Neural Information Processing Systems, 30.
> - Wu, J., Xiong, H., & Chen, J. (2010). COG: local decomposition for rare class analysis. Data Mining and Knowledge Discovery, 20, 191-220.

---

> > ### Comment · Reviewer_nZ1i · 2023-11-21
> > **Thanks for your response**
> >
> > I wanted to thank the authors for their response.
> > Thanks for adding the error bars and also for the further explanation of the method.
> > After reading other reviewers' comments, I decided I will keep my score.

---

### Meta-Review · Area_Chair_T4eY · 2023-12-12

**Metareview:**

In this paper, the authors propose a new method for improving the performance of imbalance classification by subdividing the majority class into subclasses and creating a balanced problem. The authors proposed to use clustering to divide the majority class. The reviewers have indicated the paper in its current form has several issues that should be resolved before it can be presented at a major conference:
1 Novelty: There is a paper by Wu et al. 2010 that proposes a similar idea, and this paper does not propose a significant deviation from it.
2 Clarity: the paper is not clearly explained, and it is not reproducible.
4 Results: the results are inconclusive, and there is a clear application in which this algorithm would provide a significant improvement.

**Justification For Why Not Higher Score:**

THe paper proposes a simple algorithm that it is not novel or interesting.

**Justification For Why Not Lower Score:**

not aplicable.

---

### Decision · Program_Chairs · 2024-01-16

Reject